# Implicit Stacked Autoregressive Model for Weather Forecasting

## Abstract

The escalating impact of global climate change has heightened the demand for accurate and reliable weather forecasting. As we confront challenges from extreme weather events and seek to understand long-term climate variations, precise predictions are considered hard to achieve. Autoregressive methods, commonly used for temporal modeling, have paved the way for data-driven approaches to overcome the limitations of traditional numerical methods. However, their performance significantly diminishes when tasked with extended long-term predictions due to error accumulation stemming from prior predictions. Meanwhile, the lead time embedding method has been explored as a means to mitigate the error accumulation in autoregressive models. Nevertheless, this method does not guarantee the preservation of correlations between outputs, a crucial consideration in atmospheric phenomena. To address these limitations, we present the Implicit Stacked Autoregressive Model for Weather Forecasting (IAM4WF). This innovative model combines the strengths of both autoregressive and lead time embedding methods. It offers flexibility in modeling the lead time of outputs, akin to the lead time embedding method, and it also iteratively integrates its predictions, similar to the autoregressive approach. We rigorously evaluate IAM4WF against weather and climate forecasting datasets, and additional common video frame prediction datasets. Our findings underscore IAM4WF's superior performance across six tested datasets. Result videos are best viewed on our project website: `iam4wf.github.io/project-page`.

## 1 Introduction

The atmosphere on the Earth is a chaotic system (Lorenz, 1963). This inherent chaos poses a significant challenge in accurately predicting the future states of weather phenomena, including variables such as clouds, temperature, and other relevant information. In order to capture the intricate dynamics of the atmosphere, the Numerical Weather Prediction (NWP) was introduced. NWP leverages the laws of physics to make forecasts. Despite the continuous development and widespread utilization of NWP over the decades, it encounters challenges in the aspect of computational cost and the limited understanding of the chaotic nature of complex atmospheric phenomena (Bauer et al., 2015).

Recently, the emergence of data-driven methodologies for weather forecasting demonstrated a potential to rival the operational NWP models (Lam et al., 2022; Pathak et al., 2022; Keisler, 2022; Gao et al., 2022b; Bi et al., 2023). Many studies have predominantly focused on the autoregressive architecture to achieve versatility in forecasting lead times (Lam et al., 2022; Pathak et al., 2022; Keisler, 2022). While these approaches offer weather predictions for various time scales, they inherently involve error propagation since they employ their own predictions as input information (Bi et al., 2023; Ning et al., 2023). Furthermore, they are intrinsically restricted to making predictions within the fixed time intervals for which they were trained. In an effort to broaden the scope of lead time options, several works have explored the lead time embedding method (Sønderby et al., 2020; Espeholt et al., 2022). This method takes the target lead time as input and generates predictions based on the specified lead time. However, it is important to note that the lead time embedding methods cannot guarantee dense spatiotemporal correlation of model outputs, which are fundamental properties of many meteorological phenomena.

To address these challenges, we propose a novel structured forecasting model that integrates the lead time embedding method with the autoregressive approach. Our implicit weather forecasting prediction model mimics the versatility of autoregressive models without the error-accumulated structure. In addition to this, we introduce a stacked autoregressive method to address the lack of output correlation in lead time embedding methods. Our proposed model, the Implicit Stacked Autoregressive Model for Weather Forecasting (IAM4WF), incorporates these two dominant methods of data-driven weather forecasting and achieves state-of-the-art performance on three benchmark weather and climate prediction datasets.

Our contributions can be summarized as follows:

- We propose a novel autoregressive model that predicts future frames recursively with a Multiple-In-Single-Out design that can consider the spatiotemporal correlation of weather events.
- We propose a stacked structure to avoid the problem of error accumulation and missing input data.
- We demonstrate that our Implicit Stacked Autoregressive Model for Weather Forecasting (IAM4WF) achieves state-of-the-art performance on three benchmark datasets for weather and climate prediction.

## 2 RELATED WORK

**Autoregressive Approach for weather forecasting.** Most of the efforts to employ data-driven models for weather forecasting have been dominated by the autoregressive approach. The most dominant approach **deploys memory units** (Shi et al., 2015; Ayzel et al., 2020; Kumar et al., 2019; Wang et al., 2022; Ehsani et al., 2022). Typically, this type of work deploys memory cells like Long Short-Term Memory (Graves & Graves, 2012) or Gated Recurrent Unit (Cho et al., 2014). Such an approach, however, presents a mathematical vulnerability in which errors can propagate and accumulate. It makes multi-step forecasts increasingly uncertain, particularly as the lead time expands. This challenge is not just theoretical; it has been evidenced in practical applications. For instance, Shi et al. (2015) unveiled the ConvLSTM structure with an emphasis on long-term forecasting, while Ayzel et al. (2020) merged the autoregressive technique with the U-Net structure, predicting rainfall 60 minutes into the future in 5-minute segments. Notwithstanding these applications, the issue of error accumulation is still deeply ingrained, with results deteriorating significantly for extended lead times. This has steered the recent trend away from purely autoregressive methods using memory unit.

Another development in weather forecasting models is the emergence of the autoregressive structure **without a memory cell**, which utilizes multiple past frames as input (Bi et al., 2023; Zhang et al., 2023; Pathak et al., 2022; Gao et al., 2022b). A model highlighting this trend, Bi et al. (2023), achieved leading-edge outcomes using a 3D Swin-Transformer architecture, taking cues from Liu et al. (2021). This model processes a span of 10 days with 24-hour intervals and forecasts the subsequent 10 days in similar periods. However, a discernible limitation is its inability to predict beyond its trained time frames. Therefore, for varying intervals, such as 1-hour or 12-hour predictions, separate models are necessary, and the model's performance tends to diminish for non-fixed lead times.

**Lead Time Embedding for Weather Forecasting.** A distinct approach to weather forecasting, known as the **lead time embedding** technique (Sønderby et al., 2020; Espeholt et al., 2022), represents a significant departure from the traditional autoregressive methods. The model put forth by Sønderby et al. (2020) uniquely processes both sequential spatiotemporal data and a defined lead time as inputs, generating forecasts specifically tailored to that lead time. This strategy effectively circumvents the recurring error accumulation dilemma inherent to autoregressive methods. To further enhance the predictive capabilities of this model, Espeholt et al. (2022) introduced a physics-based constraint, thereby improving the model's forecasting performance. However, a challenge persists: the model's deliberate omission of feedback from previous forecasts creates an informational void for long-term predictions. It potentially compromises model accuracy and correlation among model outputs.

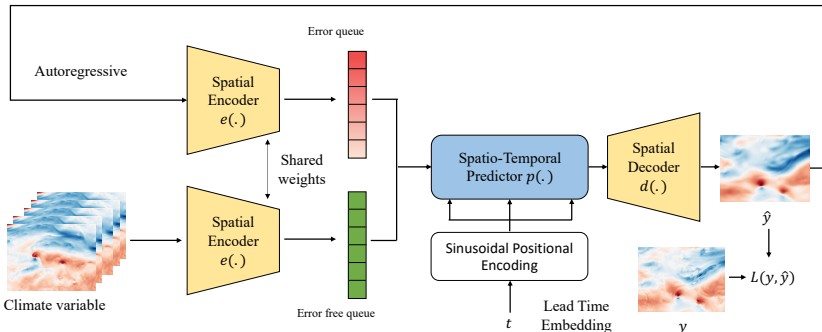

Figure 1: An overview of IAM4WF's training process. IAM4WF is an implicit model for time step $t$. IAM4WF consists of a spatial encoder $e(\cdot)$, a spatio-temporal predictor $p(\cdot)$, and a spatial decoder $d(\cdot)$. Note that, IAM4WF is trained stacked autoregressive manner with error-prone queue.

## 3 IMPLICIT STACKED AUTOREGRESSIVE MODEL

### 3.1 MODEL DESIGN

In our model framework named Implicit Stacked Autoregressive Model for Weather Forecasting (IAM4WF), the goal of the future frame prediction model $F(\cdot)$ is to map the input $X \in \mathbb{R}^{C \times T \times H \times W}$ to the target output $y_t \in \mathbb{R}^{C \times H \times W}$ using the lead time $t$ and the history of model predictions $\hat{Y}_{t-1}$. The learnable parameters of the model are denoted by $\Theta$. These parameters are optimized to minimize the following objective function:

$$\min_{\Theta} \sum_{t < \hat{T}} L(F_{\Theta}(X, \hat{Y}_{t-1}, t), y_t), \tag{1}$$

where the objective is to learn the mapping from inputs to the $\hat{T}$-time future frame $y_t$. In this context, $X = \{x_{T-1}, x_{T-2}, \ldots, x_0\}$ represents the $T$ observed frames, $\hat{Y}_t = \{\hat{y}_1, \hat{y}_2, \ldots, \hat{y}_t\}$ represents the history of $t$ predicted frames, $Y = \{y_1, y_2, \ldots, y_{\hat{T}}\}$ represents the $\hat{T}$ ground-truth future frames, $C$ is the number of channels, $T$ is the observed frame length, $\hat{T}$ is the future frame length, $H$ is the height, and $W$ is the width of the frames. We define $\hat{Y}_0$ as the empty set, with this case being an exception.

Various loss functions $L$ can be utilized for optimization, such as mean squared error (MSE) (Gao et al., 2022a), mean absolute error (MAE) (Ning et al., 2023), smooth loss (Seo et al., 2022), or perceptual loss (Shouno, 2020). In this work, we specifically employ the MSE loss, which is the most commonly used loss function for future frame prediction tasks.

**Relationship with the previous approaches** Both IAM4WF and existing autoregressive models use model outputs as subsequent inputs in succession. However, IAM4WF distinctively re-engages with all initial observations $X$, irrespective of the extent of the history of predicted frames $\hat{Y}$. In contrast, traditional autoregressive models work with either single or multiple fixed-length inputs. Over time, these models lose track of the initial observation, which leads to error accumulation, especially in long-term predictions. This key distinction positions IAM4WF as a potentially more robust framework for long-term forecasting scenarios.

In terms of the lead time embedding approach, IAM4WF addresses the error accumulation that's commonly associated with traditional autoregressive models. By using lead time $t$ as an input, models are capable of generating specific interval forecasts, effectively tackling error propagation. Unlike other lead time models, IAM4WF, with its stacked autoregressive method, ensures both adaptable lead times and the preservation of spatiotemporal correlations — vital for time-consistent prediction. In Section 4.3, we highlight the empirical advantages of our design choice.

## 3.2 Model Instantiation

Figure 1 illustrates our model instantiation of IAM4WF in this work. IAM4WF comprises multiple modules, including an encoder, error-free and error-prone queues, a predictor, and a decoder. Given an input $X$ and target time step $t$, $X$ passes through the encoder and its output is given to the error-free queue. The iterative prediction process up to the $t-1$ time step, $\hat{Y}_{t-1} = \{\hat{y}_1, \hat{y}_2, \ldots, \hat{y}_{t-1}\}$, is stacked in the error-prone queue. Subsequently, the error-free queue and error-prone queue are concatenated, and the concatenated output is passed through the predictor and decoder to produce the final output $\hat{y}_t$.

**Encoder-Predictor-Decoder Framework** In the task of video frame prediction, the current leading architectural paradigm is the *encoder-predictor-decoder* structure (Gao et al., 2022a). Unlike autoregressive models, the encoder-predictor-decoder structure is trained using convolutional neural networks (CNN) to map multi-step inputs $X$ to multi-step outputs $Y$. The *encoder* $e(\cdot)$ functions to extract features from observed frames $X$. In contrast to the *encoder*, which treats the observed frames as independent images without considering their spatiotemporal relationships, the *predictor* $p(\cdot)$ is tasked with capturing these relationships and transforming them into features for predicting future frames. Finally, the *decoder* $d(\cdot)$ reconstructs the forthcoming frames $Y$ based on the features provided by the *predictor*. IAM4WF adopts the encoder-predictor-decoder structure, which consists of (Conv, LayerNorm (Ba et al., 2016), SiLU (Elfwing et al., 2018)) for the encoder, (Conv, LayerNorm, SiLU, PixelShuffle (Shi et al., 2016)) for the decoder, and ConvNeXt (Liu et al., 2022) blocks for the predictor. In contrast to the original encoder-predictor-decoder structure, our model's output is a specific future frame corresponding to the given target lead time, rather than predicting multiple future frames simultaneously.

**Error-prone Queue & Error-free Queue** To ensure that IAM4WF retains the history of the initial observation and predictions, it incorporates two key components: an *error-free queue* and an *error-prone queue*. The error-free queue $Q_{\text{error-free}}$ serves as an explicit memory bank, storing feature vectors derived from the observed frames $X$. This explicit memory queue enables the model to preserve all the information from the initial observation when making forecasts across all lead times. In contrast, common autoregressive models undergo alterations of initial observed information as the lead time increases. In addition to this, IAM4WF introduces an error-prone queue $Q_{\text{error-prone}}$, which comprises feature vectors from the history of predicted frames $\hat{Y}$. This component explicitly maintain the history of predictions and enables the model to consider spatiotemporal correlations between predictions, which is not guaranteed in the common lead time embedding approaches.

**Lead Time Embedding** The approach in IAM4WF utilizes the lead time embedding methodology for flexible lead time prediction. For a given lead time $t$, sinusoidal positional embedding is performed at position $t$. This embedded representation, $t_{\text{embed}}$, is subsequently passed through an MLP, where its dimensionality is adjusted to match the number of channels in each layer of the predictor $p(\cdot)$.

The sinusoidal positional encoding for a given position $t$ and dimension $i$ is articulated as proposed by Vaswani et al. (2017):

$$PE_{(t,2i)} = \sin\left(\frac{t}{10000^{\frac{2i}{d}}}\right), \tag{2}$$

$$PE_{(t,2i+1)} = \cos\left(\frac{t}{10000^{\frac{2i}{d}}}\right), \tag{3}$$

where $PE$ denotes the 2D positional encoding matrix, $i$ is the dimension index, and $d$ stands for the embedding dimension. The complete positional embedding for $t$ is obtained by aggregating across the dimension $i$:

$$t_{\text{embed}} = \left[PE_{(t,0)}, PE_{(t,1)}, \ldots, PE_{(t,d-1)}\right]. \tag{4}$$

Upon computation of the embedding $t_{\text{embed}}$, it is forwarded through two fully-connected layers equipped with the GELU activation function (Hendrycks & Gimpel, 2016): (Linear, GELU, Linear). In the context of this work, the lead time embedding $s(\cdot)$ is described as a composition of the sinusoidal positional embedding and processing through a two-layer network.

---

**Algorithm 1** PyTorch-style pseudo code of loss computation for IAM4WF

---

```python
def compute_loss_IAM4WF(X, Y, hat_T):
    """
    Args:
    - X (torch.Tensor): Observed frame input (batch_size, C, T, H, W).
    - Y (torch.Tensor): Future frame target (batch_size, C, hat_T, H, W).
    - hat_T (int): Target time step.
    """
    Q_error_free = encoder(X)
    B, C, T, H, W = X.size()
    Q_error_prone = torch.zeros(B, hat_T - 1, C, H, W)

    predicted_frame = None
    total_loss = 0.0

    for t in range(hat_T):
        if t > 0:
            Q_error_prone[:, t - 1, :, :, :] = e(predicted_frame)
        stacked_features = torch.cat((Q_error_free, Q_error_prone), dim=1)
        predicted_frame = d(p(stacked_features, s(t)))
        loss = mse_loss(predicted_frame, Y[:, :, t, :, :])
        total_loss += loss
    return total_loss
```

---

## 3.3 MODEL TRAINING

The stacked autoregressive model $F(\cdot)$ is trained to predict the future frame $y_t$ by taking as input the observed frames $X$, the sequentially predicted frames $\hat{Y}_{t-1}$, and the lead time step $t$. Thus, given a pair of input frames $X$, target frames $Y$, and the final lead time $\hat{T}$, objective function of the model $F(\cdot)$ is computed as:

$$\sum_{t<\hat{T}} L\left(d\left(p(e(X), e(\hat{Y}_{t-1}), s(t))\right), y_t\right), \tag{5}$$

where $e(\cdot)$ is the encoder, $p(\cdot)$ is the predictor, $d(\cdot)$ is the decoder, and $s(\cdot)$ is the lead time embedding.

Algorithm 1 describes the loss computation for IAM4WF. The proposed model is trained sequentially, with $Q_{\text{error-prone}}$ being accumulated from $t = 0$ to $\hat{T} - 1$. It is noteworthy that $Q_{\text{error-prone}}$ is pre-allocated as a zero tensor, following a similar approach as that used in Masked Autoencoders (MAE) (He et al., 2022). This training strategy enables both the predictor $p(\cdot)$ and the decoder $d(\cdot)$ to be trained on inputs that incorporate errors from $Q_{\text{error-prone}}$ with the error-free features $Q_{\text{error-free}}$ during training. As a result, the predictions are robust against error accumulation during inference. Furthermore, this approach facilitates the consideration of correlations between outputs.

## 4 EXPERIMENTS

In this section, we present the evaluation of IAM4WF on benchmarks for both weather/climate forecasting and video frame prediction. Additionally, we conducted an ablation study to gain insights into the design of weather forecasting models. We also showcase the qualitative results of video frame interpolation, one of IAM4WF's notable strengths, in Appendix A.4.

**Benchmark Datasets**  We evaluate IAM4WF on six datasets. The SEVIR, ICAR-ENSO, and WeatherBench (Rasp et al., 2020) datasets are benchmarks for weather and climate prediction. The SEVIR dataset (Veillette et al., 2020) includes radar-derived measurements of vertically integrated liquid water (VIL) taken at 5-minute intervals with 1 km spatial resolution, making it a benchmark for rain and hail detection. The ICAR-ENSO dataset (Ham et al., 2019) merges observational and simulation data to provide forecasts of El Niño/Southern Oscillation (ENSO), a sea surface temperature (SST) anomaly in the Equatorial Pacific that is a significant predictor of global seasonal climate. Understanding and predicting global weather patterns is crucial. The WeatherBench dataset offers data at resolutions of $5.625°$ ($32 \times 64$ grids) and $1.40625°$ ($128 \times 256$ grids). In the WeatherBench-S setup, each climatic variable is trained individually on data from 2010-2015, validated in 2016, and

Table 1: Performance comparison experiment results between IAM4WF and recent leading weather prediction models on SEVIR dataset and ICRA-ENSO dataset.

| Model | SEVIR | | | | | ICAR-ENSO | | | | |
|---|---|---|---|---|---|---|---|---|---|---|
| | #Param. (M) | GFLOPS | CSI-M | CSI-160 | CSI-16 | MSE | #Param. (M) | GFLOPS | C-Niño3.4-M | C-Niño3.4-WM | MSE |
| UNet (Veillette et al., 2020) | 16.6 | 33 | 0.3593 | 0.1278 | 0.6047 | 4.1119 | 12.1 | 0.4 | 0.6926 | 2.102 | 2.868 |
| ConvLSTM (Shi et al., 2015) | 14.0 | 527 | 0.4185 | 0.2157 | 0.7441 | 3.7532 | 14.0 | 11.1 | 0.6955 | 2.107 | 2.657 |
| PredRNN (Wang et al., 2017) | 23.8 | 328 | 0.4080 | 0.2928 | 0.7569 | 3.9014 | 23.8 | 85.8 | 0.6492 | 1.910 | 3.044 |
| PhyDNet (Guen & Thome, 2020) | 3.1 | 701 | 0.3940 | 0.2767 | 0.7507 | 4.8165 | 3.1 | 5.7 | 0.6646 | 1.965 | 2.708 |
| E3D-LSTM (Wang et al., 2018b) | 12.9 | 523 | 0.4038 | 0.2708 | 0.7059 | 4.1702 | 12.9 | 99.8 | 0.7040 | 2.125 | 3.095 |
| Rainformer (Bai et al., 2022) | 19.2 | 170 | 0.3661 | 0.2675 | 0.7573 | 4.0272 | 19.2 | 1.3 | 0.7106 | 2.153 | 3.043 |
| Earthformer (Gao et al., 2022b) | 7.6 | 257 | 0.4419 | 0.3232 | 0.7513 | 3.6957 | 7.6 | 23.9 | 0.7329 | 2.259 | 2.546 |
| IAM4WF | 34.7 | 392 | **0.4607** | **0.3430** | **0.7761** | **2.9371** | 34.2 | 11.8 | **0.7698** | **2.484** | **1.563** |

Table 2: Performance comparison experiment results between IAM4WF and recent leading weather forecasting models on weather bench dataset.

| Model | Weather Bench | | | | | | | |
|---|---|---|---|---|---|---|---|---|
| | Temperature (t2m) | | Humidity (r) | | Wind (uv10) | | Cloud Cover (tcc) | |
| | MSE | RMSE | MSE | RMSE | MSE | RMSE | MSE | RMSE |
| ConvLSTM (Shi et al., 2015) | 1.521 | 1.233 | 35.146 | 5.928 | 1.8976 | 1.3775 | 0.0494 | 0.2223 |
| E3D-LSTM (Wang et al., 2018b) | 1.592 | 1.262 | 36.534 | 6.044 | 2.4111 | 1.5528 | 0.0572 | 0.2393 |
| PredRNN (Wang et al., 2017) | 1.331 | 1.154 | 37.611 | 6.133 | 1.8810 | 1.3715 | 0.0550 | 0.2346 |
| PhyDNet (Guen & Thome, 2020) | 285.9 | 16.91 | 239.00 | 15.46 | 16.798 | 4.0986 | 0.0991 | 0.3148 |
| MIM (Wang et al., 2019) | 1.784 | 1.336 | 36.534 | 6.044 | 3.1399 | 1.7720 | 0.0572 | 0.2393 |
| MetNet (Sønderby et al., 2020) | 1.545 | 1.243 | 52.199 | 7.224 | 2.0072 | 1.416 | 0.0505 | 0.2247 |
| SimVP (Gao et al., 2022a) | 1.238 | 1.113 | 34.355 | 5.861 | 1.9993 | 1.4140 | 0.0476 | 0.2182 |
| IAM4WF | **1.150** | **1.072** | **30.833** | **5.552** | **1.5842** | **1.258** | **0.0453** | **0.2128** |

tested in 2017-2018 at one-hour intervals. Conversely, WeatherBench-M trains all variables simultaneously using data from 1979-2015, with the same validation and testing periods as WeatherBench-S but at six-hour intervals.

MovingMNIST (Srivastava et al., 2015), TrafficBJ (Zhang et al., 2017), and Human 3.6 (Ionescu et al., 2013) serve as benchmark datasets for video frame prediction. MovingMNIST is composed of synthetically generated video sequences, each featuring two digits representing numbers between 0 and 9. TrafficBJ contains taxicab GPS and meteorological data recorded in Beijing. Human 3.6 provides motion capture data of a person taken using a high-speed 3D camera.

**Evaluation metric** For the evaluation of video frame prediction datasets, we adopt widely used evaluation metrics, including Mean Square Error (MSE), Mean Absolute Error (MAE), Peak Signal to Noise Ratio (PSNR), Structural Similarity Index Measure (SSIM), and Fréchet Video Distance (FVD) (Unterthiner et al., 2018). For rain forecasting models, we use the Critical Success Index (CSI) as an evaluation metric (Shi et al., 2015). In addition, we validate ENSO forecasting using the Nino SST indices (Gao et al., 2022b). Specifically, the Nino3.4 index represents the averaged SST anomalies across a specific Pacific region ($170°$-$120°$W, $5°$S-$5°$N), and defines El Niño/La Niña events based on the SST anomalies around the equator.

**Implementation details** We use the Adam optimizer (Kingma & Ba, 2014) with $\beta_1 = 0.9$ and $\beta_2 = 0.999$, and a cosine scheduler without warm-up (Loshchilov & Hutter, 2016) in all experiments. The learning rate is set to 0.001, and the mini-batch size is 16. To mitigate the learning instability of the model, we introduce an exponential moving average (EMA) approach (Rombach et al., 2022), which is not typically adopted in existing future frame prediction models. Updates to the EMA model occur every 10 iterations, starting at the 2000th iteration, with an update momentum set to 0.995. The MovingMNIST dataset is trained for 10K epochs, while all other datasets undergo training for 2K epochs.

## 4.1 Weather/Climate Benchmark Results

**SEVIR** The SEVIR data possesses a relatively higher resolution, and water objects' shapes in it are less distinct compared to numbers and humans in other datasets. Even though Earthformer already set a state-of-the-art benchmark on this dataset, our model surpassed its scores with a CSI-M of **0.4607** and MSE of **2.9371**, as depicted in Tab. 1. The CSI index typically validates the model's precipitation forecast accuracy (Jolliffe & Stephenson, 2012), defined as CSI =

Table 3: Performance comparison results of IAM4WF and recent leading approaches on three future frame prediction video frame prediction datasets. IAM4WF achieved state-of-the-art performance on all three benchmark datasets, which have different characteristics: MovingMNIST, TrafficBJ, and Human 3.6. The asterisk (*) indicates the performance reported in the author's paper.

| | MovingMNIST | | | | TrafficBJ | | | | Human 3.6 | | | |
|---|---|---|---|---|---|---|---|---|---|---|---|---|
| Method | MSE | MAE | SSIM | FVD | MSE ×100 | MAE | SSIM | FVD | MSE/10 | MAE /100 | SSIM | FVD |
| ConvLSTM (Shi et al., 2015) | 103.3 | 182.9 | 0.707 | 102.43 | 48.5 | 17.7 | 0.978 | 133.28 | 50.4 | 18.9 | 0.776 | 153.90 |
| PredRNN (Wang et al., 2017) | 56.8 | 126.1 | 0.867 | 152.34 | 46.4 | 17.1 | 0.971 | 113.06 | 48.4 | 18.9 | 0.781 | 102.19 |
| Causal LSTM (Wang et al., 2018a) | 46.5 | 106.8 | 0.898 | 86.39 | 44.8 | 16.9 | 0.977 | 55.20 | 45.8 | 17.2 | 0.851 | 66.38 |
| MIM (Wang et al., 2019) | 44.2 | 101.1 | 0.910 | 178.0 | 42.9 | 16.6 | 0.971 | 153.29 | 42.9 | 17.8 | 0.790 | 122.46 |
| E3D-LSTM (Wang et al., 2018b) | 41.3 | 86.4 | 0.920 | 22.20 | 43.2 | 16.9 | 0.979 | 90.21 | 46.4 | 16.6 | 0.869 | 88.91 |
| PhyDNet (Guen & Thome, 2020) | 24.4 | 70.3 | 0.947 | 15.87 | 41.9 | 16.2 | 0.982 | 50.76 | 36.9 | 16.2 | 0.901 | 80.22 |
| SimVP (Gao et al., 2022a) | 23.8 | 68.9 | 0.948 | 11.75 | 41.4 | 16.2 | 0.982 | 13.08 | 31.6 | 15.1 | 0.904 | 9.52 |
| MIMO-VP* (Ning et al., 2023) | 17.7 | 51.6 | 0.964 | - | - | - | - | - | - | - | - | - |
| IAM4WF | 16.9 | 49.9 | 0.963 | 7.319 | 39.0 | 16.7 | 0.985 | 8.251 | 12.9 | 11.5 | 0.948 | 5.251 |

Hits/(Hits + Misses + F.Alarms). Both CSI-160 and CSI-16 calculate Hits ($\text{obs} = 1, \text{pred} = 1$), Misses ($\text{obs} = 1, \text{pred} = 0$), and False Alarms ($\text{obs} = 0, \text{pred} = 1$) based on binary thresholds from $[0, 255]$ pixel values. We employed thresholds $\{16, 74, 133, 160, 181, 219\}$, and the CSI-M represents their average value (Gao et al., 2022b).

**ICAR-ENSO** The forecast evaluation metric, termed C-Nino3.4, computes the correlation skill of the three-month-averaged Nino3.4 index (Ham et al., 2019). We make forecasts for up to 14-month SST anomalies (extending 2 months beyond the input data used for three-month averaging) based on a 12-month SST anomaly observation. As Table 1 indicates, IAM4WF outperforms all other methods across the board. The metric C-Nino3.4-M represents the mean correlation skill over 12 forecasting steps, while C-Nino3.4-WM is its time-weighted average. We also assessed MSE to gauge the spatiotemporal accuracy between predicted and observed values. Our model not only enhanced the C-Nino3.4 indices but notably reduced the MSE by more than 1 compared to other models.

**Weather Bench** Tab. 2 showcases the results from an experiment where IAM4WF was assessed on Weather Bench, a task dedicated to weather prediction. We employed baseline models typically utilized for video prediction tasks. As seen in Table 2, our IAM4WF demonstrates state-of-the-art performance across weather variables, including Temperature (t2m), Humidity (R), Wind (uv10), and Cloud Cover (tcc). These findings suggest that IAM4WF holds significant promise for global modeling tasks.

## 4.2    VIDEO FRAME PREDICTION RESULTS

**MovingMNIST** In Tab. 3, the first row illustrates the performance of IAM4WF in comparison to other recent leading approaches on the MovingMNIST dataset. Both PhyDNet and SimVP are acknowledged for their state-of-the-art achievements. Notably, PhyDNet employs a memory cell, while SimVP does not incorporate one. On the MovingMNIST dataset, IAM4WF registers an MSE of **17.3**, MAE of **49.9**, SSIM of **0.963**, and FVD of **7.319**, surpassing other state-of-the-art models for this rule-based synthetic dataset.

**TrafficBJ** The second row of Tab. 3 showcases IAM4WF's performance in comparison to other methods on the TrafficBJ dataset. Given the predominantly linear relationship between the past and future in the TrafficBJ dataset, most models exhibit saturated values for metrics like MSE, MAE, SSIM, and FVD. However, IAM4WF still achieves state-of-the-art results, outdoing other methods by a significant margin. This suggests that IAM4WF is adept at handling simple real-world datasets, even those with straightforward linear relationships.

**Human 3.6** Tab. 3's third row highlights IAM4WF's performance on the Human 3.6 dataset. The Human 3.6 dataset presents a challenge due to the non-linear relationship between past and future, given its focus on predicting human behavior. The table reveals that IAM4WF excels over previous models, boasting a notable performance improvement of 19.0 in the MSE metric. Such results underscore IAM4WF's robustness when dealing with complex real-world datasets demanding non-linear modeling.

## 4.3 ABLATION STUDY

**Component effect** The results of this study are presented in Tab. 4, where we assessed the performance of IAM4WF by incrementally incorporating each component into the SimVP baseline (Gao et al., 2022a). In SimVP, we observed a performance improvement of 0.5 points by modifying the encoder structure from (Conv, GroupNorm (Wu & He, 2018), LeakyReLU (Xu et al., 2015)) to (Conv, LayerNorm, SiLU) and the decoder structure from (ConvTranspose, GroupNorm, LeakyReLU) to (Conv, LayerNorm, SiLU, PixelShuffle).

Table 4: Ablations for IAM4WF components.

| Component | #Param (M) | MSE |
|---|---|---|
| SimVP | 20.4 | 23.5 |
| +Improved Autoencoder | 20.4 | 23.0 |
| +ConvNeXt (replace predictor) | 20.4 | 18.2 |
| +Time Step MLP Embedding | 20.6 | 18.0 |
| +Stacked Autoregressive | 20.7 | 16.9 |

Furthermore, a significant performance boost of +4.8 points was achieved by replacing the existing inception block with the ConvNeXt block as the spatiotemporal predictor.

These experimental results indicate the importance of the spatiotemporal predictor in the video frame prediction task. Subsequently, by transitioning from the multi-step input and multi-step output to the lead time embedding method (multi-step input and the single-step output), we attained an additional performance gain of +0.2 points. While this improvement may appear modest, it was a pivotal step in implementing the stacked autoregressive model structure. Finally, the introduction of the stacked autoregressive method led to a substantial performance improvement of 1.1 points. These findings highlight the significant impact of the stacked autoregressive approach in enhancing overall performance.

**Output condition dependency** We conducted an ablation study on the MovingMNIST dataset to assess the impact on output conditions of IAM4WF. The results are presented in Figure 2.

The left graph contrasts the performance of SimVP and IAM4WF based on output length. As depicted, SimVP's error increases linearly with the output length, whereas IAM4WF remains more consistent, highlighting its robustness to changes in output length. The right graph offers an analysis based on the size of the time interval for both input and output; for both methods, errors rise as the time interval grows, aligning with expectations. IAM4WF shows superior performance compared to SimVP for all time interval values.

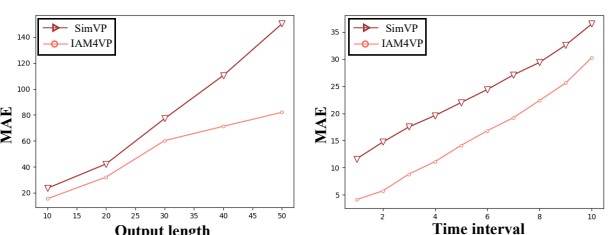

Figure 2: Performance comparison experiment according to the output length (Left) and time interval (Right) changes of SimVP and IAM4WF on the MovingMNIST dataset.

These findings underscore the importance of the IAM4WF's design, and our model can be effectively applied in more challenging tasks.

## 4.4 QUALITATIVE RESULTS

Due to page limitations, a more exhaustive qualitative video results can be found in our project website (`iam4wf.github.io/project-page`).

**ICRA-ENSO** The predicted results for the ICAR-ENSO dataset are depicted in Figure 3, showcasing the global map of SST anomalies. In the color bar, blue denotes negative SST anomalies, while red represents positive ones. The results indicate that IAM4WF tends to overestimate values but successfully predicts anomalous patterns transitioning from negative to positive, in line with observations (GT). Notably, even if the model produces incorrect predictions in the initial stages for certain regions, subsequent predictions align with the correct SST anomaly pattern. This suggests that the IAM4WF model does not carry forward early prediction errors to subsequent predictions.

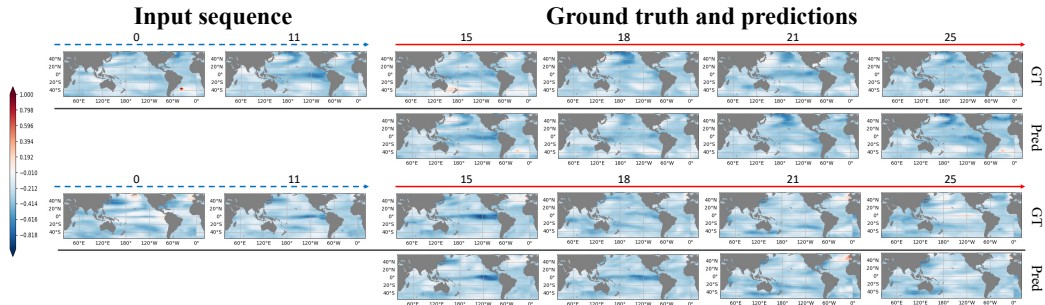

Figure 3: Prediction results of IAM4WF on the ICRA-ENSO dataset. The color bar means SST anomalies on the global map. Best viewed with zoom.

**SEVIR** Figure 4 displays the predicted outcomes of IAM4WF and Earthformer on the SEVIR dataset. This dataset signifies radar-estimated liquid water in a vertical air column, with the color bar representing water mass per unit area.

A higher VIL value is indicative of elevated water content, increasing the likelihood of precipitation or even the occurrence of hail during storms. From 12 input images (the two on the left), we forecast 12 future frames (four samples on the right). In both examples, the initial future frame predictions by both models seem plausible. However, disparities begin to emerge from frame 17 onward. Earthformer exhibits issues with blurriness, characterized by oversimplified cell boundaries and the omission of smaller cells. This is a rec-

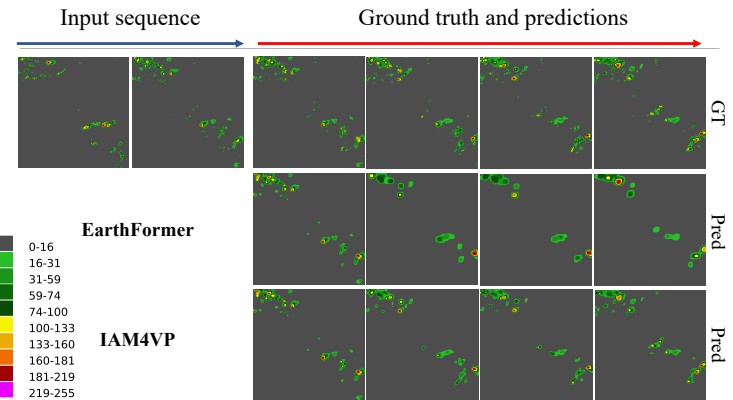

Figure 4: Prediction results of IAM4WF and Earthformer on the SEVIR dataset, represented by vertically integrated liquid water contents (0-255 scale) shown on the color bar.

ognized limitation of deep-learning-based weather prediction models. Conversely, IAM4WF offers precise predictions of cell boundaries up to frame 24. Notably, even the smaller cells are retained throughout training, suggesting that some cells, which are not apparent in Earthformer's outputs, are discernible in our model's predictions.

## 5 CONCLUSION

In this paper, we introduced a novel implicit weather forecasting model, IAM4WF, which effectively harnesses the strengths of both the autoregressive method and the lead time embedding method. By integrating the lead time embedding approach and maintaining an error-free queue, our model effectively mitigates the error accumulation problem that has hindered existing autoregressive methods. Furthermore, the incorporation of a stacked autoregressive architecture and an error-aware queue ensures that IAM4WF consistently captures the essential spatiotemporal correlations between predictions — a fundamental property in atmospheric phenomena that was often overlooked in previous lead time embedding methods. Our extensive experiments have demonstrated that IAM4WF achieves state-of-the-art performance on three weather and climate forecasting datasets, as well as three video frame prediction tasks.

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

## A APPENDIX

### A.1 IMPLEMENTATION DETAILS

IAM4WF consists of an encoder $e(.)$, a spatial-temporal predictor $p(.)$, a spatial decoder $d(.)$, and a positional encoder $s(.)$. We have empirically confirmed that the amount of computation (Flops) and accuracy depending on the number of $e(.)$, $d(.)$, and $p(.)$. Table 5 shows the number of channels and layers for each dataset. Most of the hyperparameters were set according to SimVP's Gao et al. (2022a) recipe. We empirically found that for larger image sizes, it is more efficient to use more parameters for both th encoders and decoders, while for smaller image sizes, it is more efficient to reduce the number of parameters in both the encoders and decoders.

|  | MovingMNIST | TrafficBJ | Human 3.6 | SEVIR | ICRA-ENSO |
|---|---|---|---|---|---|
| $e(.)_{channels}$ | 64 | 64 | 64 | 64 | 64 |
| $d(.)_{channels}$ | 64 | 64 | 64 | 64 | 64 |
| $p(.)_{channels}$ | 512 | 256 | 128 | 512 | 128 |
| $e(.)_{num}$ | 4 | 3 | 4 | 5 | 2 |
| $d(.)_{num}$ | 4 | 3 | 4 | 5 | 2 |
| $p(.)_{num}$ | 6 | 4 | 8 | 6 | 4 |

Table 5: IAM4WF hyperparameter settings of each component.

### A.2 QUALITATIVE COMPARISON RESULTS

Figure 5 presents a qualitative comparison between competitive models and IAM4WF. As shown in the Figure 5, representative autoregressive models such as ConvLSTM (Shi et al., 2015), Pre-dRNN (Wang et al., 2017), and E3D-LSTM (Wang et al., 2018b), which are commonly used in video frame prediction, exhibit accumulated errors and produce blurry images as the lead time increases. In contrast, both SimVP and IAM4WF produce relatively clear images across all lead times, with IAM4WF yielding the clearest images among them. It is worth noting that SimVP here does not make use of its own prediction as input because the target lead time does not extend beyond the output length of the trained model.

### A.3 DOES IAM4WF REALLY CONSIDER THE CORRELATION BETWEEN EACH LEAD TIME OUTPUT?

IAM4WF utilizes an error-prone queue, which contains features of its own predicted frames, for inference. To investigate whether IAM4WF depends solely on the error-free queue (composed of features of observed frames) for information during inference, we conducted qualitative experiments. Figure 6 illustrates the outputs of IAM4WF for different error-prone queue configurations. As seen in the results of the *Random* experiment in Figure 6, randomly shuffling the error-prone queue leads to inaccurate IAM4WF outputs. This experiment result indicates that IAM4WF is affected by the error-prone queue. Additionally, the best performance was achieved when the error-prone queue is stacked in the correct order (*Original*). However, as demonstrated in the *All Zero* experiment, IAM4WF is still capable of generating plausible images even when the error-prone queue is not used.

### A.4 TIME INTERPOLATION FOR VIDEO PREDICTION

The target lead time is provided as input to IAM4WF, enabling the model to make predictions at arbitrary time intervals. As a result, IAM4WF has the flexibility to forecast future frames at time points that were not explicitly observed during training. Figure 7 shows the qualitative results of time interpolation using the MovingMNIST Dataset. These results demonstrate that IAM4WF can perform video frame interpolation, a task that existing video frame prediction models were unable to achieve. The ability of IAM4WF to make predictions at arbitrary time points is particularly advantageous in weather and climate forecasting Interpolated prediction enables the model to make

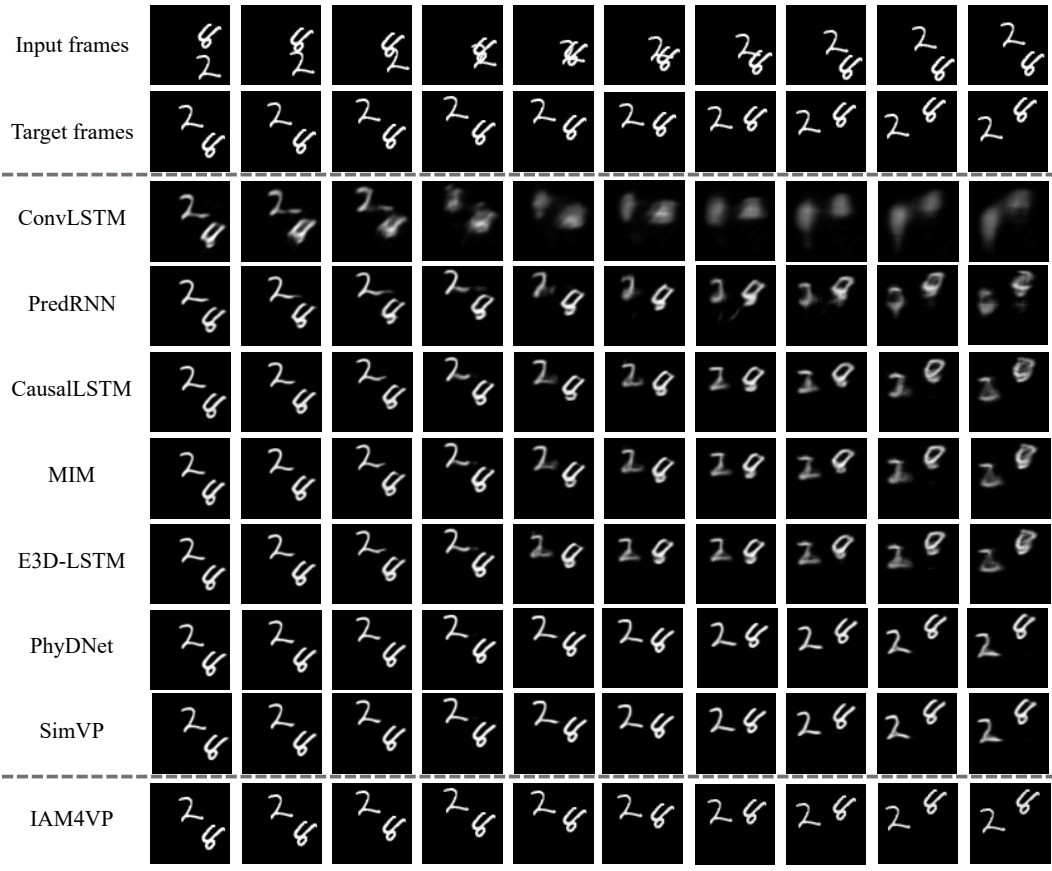

Figure 5: The qualitative comparison between competitive models and IAM4WF on the MovingM-NIST dataset.

predictions within shorter intervals compared to the actual acquired data, enhancing its temporal resolution. Furthermore, IAM4WF's capability to handle arbitrary time intervals opens up opportunities to exploit data from multiple satellites, each with its own repeat cycle. This flexibility addresses the challenges of integrating information from various sources with different temporal characteristics, which is essential for improving the accuracy of weather and climate forecasting models.

## B  MOTIVATION OF MULTI-INPUT-MULTI-OUTPUT ARCHITECTURE

In the context of a video frame prediction task, models can be categorized based on their input and output types. Recurrent neural network-based architecture such as PredRNN and ConvLSTM can be defined as Single-Input and Single-Output (SISO) architectures. In these models, the input for one forward pass of the model is a single frame, and the output is also a single frame. On the other hand, models like SimVP, which is a state-of-the-art architecture in video frame prediction, can be categorized as Multi-Input and Multi-Output (MIMO) architectures. These models receive the entire video frame as input and predict multiple future frames simultaneously (Gao et al., 2022a; Ning et al., 2023). In this section, we aim to investigate the significant factors in MIMO architectures that influence performance.

### B.1  ANALYSIS

Recent studies suggest that the MIMO model, may provide an advantage in video prediction problems. Our questions are whether a MIMO architecture is essential for achieving high performance

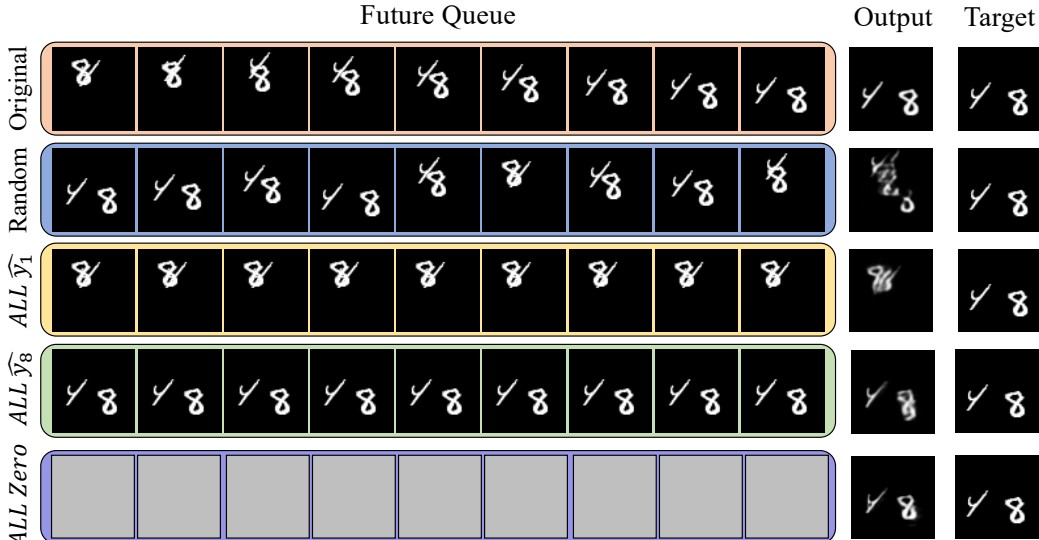

Figure 6: Qualitative comparison results of IAM4WF output according to error-prone queue configuration. *Original* represents stacking the error-prone queue in the correct order, *Random* represents random shuffling of the error-prone queue, *All* $\hat{y}_1$ and *All* $\hat{y}_8$ indicate using only $\hat{y}_1$ and $\hat{y}_8$, respectively, for the entire error-prone queue, and *All zero* represents not using the error-prone queue at all. Note that the value of $t$ is set to 9.

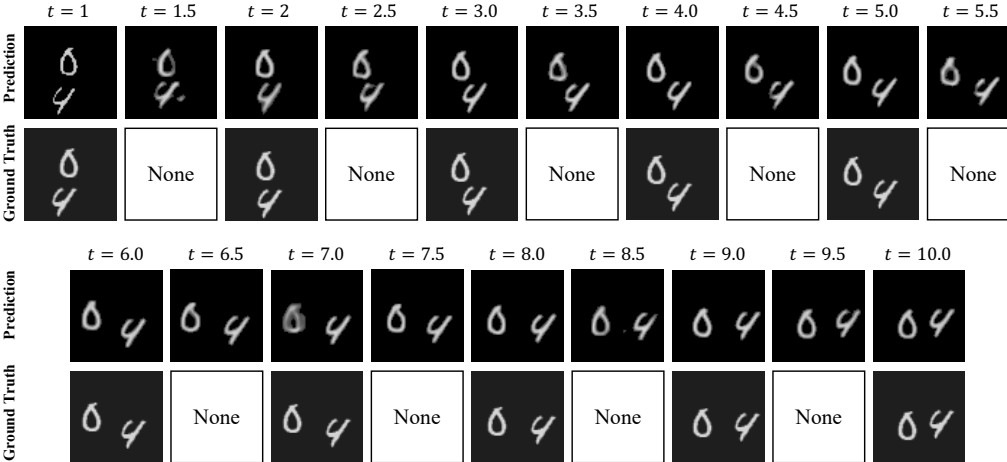

Figure 7: Future frame prediction and interpolation quantitative experiment results on the MovingMNIST dataset. Since IAM4WF is an implicit model, it was trained with a lead time interval of 1, but it can operate at an interval of 0.5 during inference.

in video prediction and whether the multiple-output component is critical in the MIMO architecture. To answer these questions, we compared autoregressive and non-autoregressive models with the same structure and hyperparameter settings. We modified the SimVP MIMO model to create the MISO-autoregressive and MISO-Multi Model and conducted ablation experiments to confirm their efficacy.

Note that the MISO-autoregressive model predicts the future frame $t_{n+1}$ by inputting time step points $B = t_0, t_1, ..., t_n$ and removing them from the queue in a first-in-first-out (FIFO) manner in the next step. In addition, the MISO-Multi Model also uses multiple inputs like the MIMO model

but models $f_{n+1}(B), f_{n+2}(B), ..., f_{n+m}(B)$, where $m$ is the number of target future frames that are specifically configured.

Table 6 presents the effectiveness comparison of the MIMO, MISO-autoregressive, and MISO-Multi Model. Our experiments reveal that Mean Squared Error (MSE) decreases as the time step increases in all experiments. Moreover, the Multi Model exhibits the highest performance. These experimental results demonstrate that the multi-input components, rather than the multi-output, significantly influence performance in the MIMO method.

Furthermore, the autoregressive model exhibits the lowest performance, indicating that the error accumulation problem is critical in future frame prediction tasks. Hence, when designing a model for future frame prediction, it is essential to consider the design of a multi-input structure that does not accumulate errors.

| Method | #Param. (M) | Training Epoch | Time Step | | | | | | | | | | MSE |
|---|---|---|---|---|---|---|---|---|---|---|---|---|---|
| | | | 1 | 2 | 3 | 4 | 5 | 6 | 7 | 8 | 9 | 10 | |
| SimVP-S (MIMO) | 20.4 | 2K | 11.6 | 14.7 | 17.5 | 19.6 | 22.0 | 24.4 | 27.1 | 29.4 | 32.6 | 36.5 | 23.5 |
| SimVP-S (MIMO)*10 | 20.4 | 20K | 8.8 | 11.9 | 15.6 | 17.3 | 19.2 | 21.4 | 25.7 | 28.8 | 30.3 | 35.2 | 21.4 |
| SimVP-L (MIMO) | 53.5 | 2K | 13.7 | 17.5 | 18.1 | 20.2 | 24.1 | 26.6 | 29.8 | 33.1 | 36.5 | 38.2 | 25.7 |
| SimVP-S (MISO-Multi Model) | 20.4*10 | 20K | **8.3** | **10.9** | **13.1** | **15.6** | **17.8** | **20.0** | **22.4** | **24.5** | **26.1** | **28.7** | **18.7** |
| SimVP-S (MISO-Autoregressive) | 20.4 | 2K | 8.3 | 13.4 | 19.2 | 24.5 | 30.3 | 36.2 | 42.6 | 48.7 | 55.2 | 62.3 | 34.1 |

Table 6: The result of comparing the efficiency of each method after changing the previous architecture SimVP model to Multiple-In-Single-Out Multi Model (MISO-Multi Model) and Multiple-In-Single-Out-autoregressive (MISO-Autoregressive) structure. All experiments were conducted on the Moving MNIST dataset.

## B.2 MOTIVATION

The initial experiments in Section B.1 suggest that the MISO model has the potential for high performance in video prediction problems. However, the MISO-Multi Model is computationally inefficient since it requires learning and inference of multiple models. Additionally, the MISO-Multi Model lacks modeling of time dependence since it does not use previous or subsequent predicted frames when inferring each future timestamp. In contrast, the MISO-Autoregressive model models time dependence but suffers from long-term error accumulation, resulting in performance degradation.

Acknowledging the potential observed in MISO models for video prediction, we designed our implicit stacked autoregressive architecture by drawing inspiration from this observation. Specifically, we integrate the error-free and error-prone queues into our framework, aiming to address the issue of long-term error accumulation that is often encountered in autoregressive models. This strategic incorporation allowed us to mitigate the challenges associated with extended predictions and enhance the overall performance of our model.

## C LIMITATION AND FUTURE WORK

Unlike the existing autoregressive model, the stacked autoregressive method lacks flexibility for lead time. For example, the lead time of the existing autoregressive model can be increased as much as desired by adjusting the autoregressive step even if the performance is degraded. However, since the stacked autoregressive method has to fix the length of the feature map input to the model, the length of the lead time cannot exceed the existing fixed length. However, the stacked autoregressive method can also perform inference in the same way as the existing autoregressive methods, but this does not match the motivation of the stacked autoregressive method. In our future work, we will conduct research on increasing the output length flexibility of the stacked autoregressive method.

