# OpenReview forum: "IMPLICIT STACKED AUTOREGRESSIVE MODEL FOR WEATHER FORECASTING"
_ICLR.cc/2024/Conference — Submitted to ICLR 2024_

### Official Review · Reviewer_GNg4 · 2023-10-20

**Soundness:** 2 fair
**Presentation:** 2 fair
**Contribution:** 1 poor
**Rating:** 3
**Confidence:** 4

**Summary:**

This paper addresses the challenges in weather forecasting due to the chaotic nature of atmospheric phenomena. It aims to overcome the limitations of existing autoregressive and lead time embedding methods used in weather forecasting. IAM4WF integrates the lead time embedding method with the autoregressive approach. It aims to offer flexibility in modeling the lead time of outputs and iteratively integrates its predictions to mitigate error accumulation and preserve correlations between outputs.

IAM4WF is designed to predict future frames recursively, considering the spatiotemporal correlation of weather events. A stacked structure is proposed to avoid error accumulation and missing input data. It adopts an encoder-predictor-decoder structure and incorporates error-prone and error-free queues to retain the history of initial observations and predictions.

IAM4WF is evaluated against benchmark datasets, demonstrating state-of-the-art performance in weather and climate prediction against several existing models and approaches.

**Strengths:**

1. The experimental results on the selected weather forecasting datasets seem promising.
2. The idea of incorporating lead time and future predicted points to inference the next future point makes sense.
3. The error-prone and error-free queues are help store information about observed frames and predicted frames.

**Weaknesses:**

1. The innovation is trivial and is not novel at all. All the concepts of memory bank, encoder-decoder structure and lead time embedding are directly from previous studies. In fact, all these concepts refer to old practices in the classical paper of Transformer (https://arxiv.org/pdf/1706.03762.pdf): memory bank is the memory mechanism practiced by RNN & LSTM line of papers, the architecture including the encoder-decoder framework directly inherits from the transformer practice, and lead time embedding is an extension of positional encoding. It is surely okay to use methods from previous papers as long as it is effective, but I hardly think it has anything to do with the weather forecasting task specifically, which is weird. It is okay to propose this model as a general framework, but it would be strange to use the same model (https://arxiv.org/pdf/2303.07849.pdf) to write multiple papers for video, weather, and potentially many other tasks: which objective is the paper really about? Without either technical innovation (substantially new model design for general deep learning framework) or application innovation (model design with in-depth adaption to application, real-world problem definition and problem-solving, real-world setting and data), this paper can hardly position itself. All the data and setting in this paper directly comes from Earthformer, which is a pioneering paper to frame the weather forecasting as a video generation task. Compared to this contribution of Earthformer, this paper only excels in several (not all, since some results are not shown) experiments (please refer to the weakness 2.) following the already defined task.

2. The experimental comparison is quite confusing. The Earthformer paper is certainly worth paying tribute to, but the setting in that paper regarding the video generation and MNIST dataset has already been criticized since it makes the objective unclear: is the paper about weather forecasting or is it about video generation? If it is about weather forecasting, does MNIST dataset matter at all? It even makes the paper less professional about the weather forecasting task. The correct way to do it should be saving everything about video experiments in the appendix, since it has nothing to do with weather forecasting! I would suggest the authors focus more on the benchmark datasets used in real-world weather forecasting nowadays, such as the advanced ECMWF simulation and NWP results (e.g., Operational IFS). Like it or not, NWP is still the main force in the area of weather forecasting. The so-called SOTA results are meaningless if they cannot compete with NWP, or at least be carried out with real-world data settings (long-range prediction, medium-range forecasting, many weather variables to consider and forecast simultaneously, etc.). I would recommend the authors to read papers such as Pangu-Weather and FourcastNet, paying extra attention on how experiments are carried out in their papers (evaluation metrics, data, problem definition, experimental setting, relation to NWP, etc.). Some datasets (excluding WeatherBench) are simply not in the interests of physicists and meteorologists, since short-term forecasting is already quite accurate with existing methods.

3. The baselines compared are too obsolete. Please consider similar settings such as Pangu-Weather, NowcastNet and FourcastNet, or at least more advanced deep learning architectures such as Fourier Neural Operator, Neural ODE, recent/strong Spatio-Temporal/video models. The experimental results for MIMO-VP are represented by "--" in TABLE 3 without explanation. The compared baselines are not consistent in TABLES 1 and 2. At least for TABLE 2, Earthformer (it is the only strong baseline) should not be scratched. If the paper is about weather forecasting, it is important to compare with advanced baselines, unless the paper makes *substantial* innovation on the technical side (at least having something new in the literature) as a general framework.

4. The presentation need substantial improvement. First, the description of neural computation is not enough. How is the spatial encoder computed and formulated? How is spatial predictor computed? Why are they called “spatial”? Second, in the experiment, how are forecasting steps set for each dataset? How is the MSE calculated: is it the average of the multiple forecasting steps? What are the GPUs used to carry out the experiments? Any reason why Temperature (t2m) Humidity (r) Wind (uv10) Cloud Cover (tcc) are selected? Is the average performance of all weather factors considered?

**Questions:**

I do not have extra questions. Please address my concerns listed in the weaknesses.

Do not get me wrong: the reason to not ignore the rise of models such as Pangu-Weather and FourcastNet is similar to LLM for NLP. It is less interesting nowadays to see the claim of a so-called SOTA performance if these successful models and even NWP are ignored. It is the authors' responsibility to persuade readers why your task and/or problem setting is still important and worth exploring. It is hard to believe this paper is important, if it cannot solve any unique challenges or meaningful existing challenges better. I suggest that, to say the least, the authors re-state the problem description and weather forecasting background, coming up with better motivations and stories. Ideally, the authors should also improve the novelty *substantially*. To say the least, giving us more sense, such as why memory mechanism is important for weather forecasting and why you choose the current architecture and how does it substantially differentiate from previous studies.

---

### Official Review · Reviewer_Dqwm · 2023-10-25

**Soundness:** 3 good
**Presentation:** 3 good
**Contribution:** 1 poor
**Rating:** 3
**Confidence:** 4

**Summary:**

The paper proposed an end-to-end framework, namely 'Implicit Stacked Autoregressive Model for Weather Forecasting' (IAM4WF), to address the challenges in long-term weather forecasting problems.

The proposed IAM4WF outperforms existing baselines on both weather forecasting datasets and video frame prediction datasets in lower forecasting errors. IAM4WF also shows stable and better forecasting results in long-term forecasting than SimVP.

**Strengths:**

1. The paper is generally well-written, and the proposed challenges in weather forecasting are well-motivated.
2. The description of the proposed methodology is clear and easy to follow.
3. The evaluations of the proposed methodology are comprehensive, and showed consistent outperformance over multiple baselines on multiple datasets.

**Weaknesses:**

1. The story-telling of the paper is problematic. The paper states it particularly focuses on the challenges of weather forecasting, and the proposed framework is also for weather forecasting. However, the paper suddenly involves video frame prediction without any context in either the Introduction or Related Work sections. Also, involving results of video frame prediction is also away from the paper's title and focuses on weather forecasting. The large existence of video frame prediction destroys the good presentation of the weather forecasting problem and confuses the reader.

2. The contribution of the work is limited, or at least not well-illustrated. The technical description is only about 1 page (with most content from existing works). The proposed framework simply follows the existing works, say encoder-predictor-decoder and positional encoding, without new innovation or interpretation. The Error-prone Queue & Error-free Queue seem to be relatively novel part of the framework, but if so, the emphasis and interpretation is very limited.

**Questions:**

1. In the framework diagram, there are shared weights between two queues, are the weights fixed or learnable? If fixed, how to set them?

2. Why does WeatherBench use different baselines than SEVIR and ICAR-ENSO, are there specific reasons to use SimVP (designed for video prediction) than Earthformer (designed for earth system forecasting)

3. For the evaluation, how many steps do all models forecast/predict?

4. In Table 3, why MAE on TrafficBJ with IAM4WF is bold, say SimVP shows a lower MAE?

5. For Figure 2 left, when output length increases, does the MAE measured based on incremental prediction, or the overall predictions?

---

### Official Review · Reviewer_sYk9 · 2023-10-31

**Soundness:** 2 fair
**Presentation:** 2 fair
**Contribution:** 2 fair
**Rating:** 3
**Confidence:** 4

**Summary:**

The paper proposes a new method for weather forecasting, namely IAM4WF, which aims to combine the strength of autoregressive models and lead-time conditioning models. The main idea of the paper is to predict the future frame using the initial observations, previously predicted frames, and the lead time embedding. IAM4WF achieves this by concatenating the features of the initial observations and the predicted frames in the channel dimension. Experiments show that IAM4WF performs better than the baselines in weather forecasting and video frame prediction.

**Strengths:**

- The paper is relevant to deep learning for weather and climate community.
- To the best of my knowledge, the idea of using both initial observations and previously predicted frames as inputs is original.

**Weaknesses:**

### Presentation
Overall, I think the presentation of the paper can be improved a lot. The current writing is unnecessarily complicated in certain sections, while missing some important details.
- The introduction of Error-prone Queue & Error-free Queue is confusing. At first, I thought the model would employ a special mechanism to handle the initial observations and predicted frames differently because of the introduction of these components. However, I learned that the model simply concatenates the observations and predicted frames over the channel dimension, treating them equally. I think introducing these terms is not necessary and removing them would make the paper more understandable.
- Weatherbench-S and Weatherbench-M are introduced but the authors did not mention what version and what resolution they use for benchmarking their model.
- In Table 2, the lead time (how far the model is predicting) is not mentioned.
- In the Component effect section, the dataset used for this experiment is not specified.
- In Table 4, +Stacked Autoregressive is not well explained. Is it equivalent to using initial observations as additional inputs?

In addition, some statements in the paper are vague and not justified or supported by experiments
- The lead time embedding methods cannot guarantee dense spatiotemporal correlation of model outputs --> What do the authors mean by  "dense spatiotemporal correlation of model outputs", and why the lead time embedding methods cannot guarantee this?
- By using lead time t as an input, models are capable of generating specific interval forecasts, effectively tackling error propagation --> This is unjustified. Conditioning on the lead time improves the flexibility of the model as it can predict the future at different temporal scales, but since it still uses its own prediction as input, it still has the problem of error propagation.
- Furthermore, this approach facilitates the consideration of correlations between outputs --> What do the authors mean by this?

### Significance
My biggest concern is regarding the significance of the proposed method. It is a minor improvement where we simply concatenate the initial observations and the predicted results over the channel dimension, and the authors have only evaluated the models on rather toy-scale datasets and benchmarks.
- In Weatherbench, the authors said the baseline models are typically utilized for video prediction tasks. I found this to be strange. Since the task is weather forecasting, the authors should have compared with state-of-the-art models for weather forecasting, not video prediction. There have been significant advancements in recent years, including FourCastNet [1], ClimaX [2], GNN [3], PanguWeather [4], Graphcast [15], etc., but the paper did not compare with any of these models. Weatherbench is a standard benchmark for weather forecasting, and this should be the main result of the paper.
- The experiments are spread out between weather forecasting and video prediction. While the main focus of the paper is on weather forecasting (given the title and the main text), the experiments seem to put more weight on the video prediction task. This is confusing, because even though there are certain similarities, these 2 tasks are very different. Weather forecasting involves a lot of atmospheric variables compared to RGB channels in videos, and these variables have complex spatiotemporal relationships that are not present in videos. It would be better if the paper could focus on weather forecasting or weather-related tasks and show the model's effectiveness in that setting only.

[1] Pathak, Jaideep, et al. "Fourcastnet: A global data-driven high-resolution weather model using adaptive fourier neural operators." arXiv preprint arXiv:2202.11214 (2022).

[2] Nguyen, Tung, et al. "ClimaX: A foundation model for weather and climate." arXiv preprint arXiv:2301.10343 (2023).

[3] Keisler, Ryan. "Forecasting global weather with graph neural networks." arXiv preprint arXiv:2202.07575 (2022).

[4] Bi, Kaifeng, et al. "Pangu-weather: A 3d high-resolution model for fast and accurate global weather forecast." arXiv preprint arXiv:2211.02556 (2022).

[5] Lam, Remi, et al. "GraphCast: Learning skillful medium-range global weather forecasting." arXiv preprint arXiv:2212.12794 (2022).

**Questions:**

- What is the reason for conditioning on lead time? The main idea of the paper is to use the initial observations to improve the autoregressive prediction of the model, and the model can achieve this without the lead time embedding component.
- Can the authors justify the use of lead time embedding? For example, the authors can show an ablation study where removing this component hurts the model's performance.

---

### Official Review · Reviewer_33K8 · 2023-11-04

**Soundness:** 2 fair
**Presentation:** 2 fair
**Contribution:** 2 fair
**Rating:** 5
**Confidence:** 4

**Summary:**

The paper presents a new model called the Implicit Stacked Autoregressive Model for Weather Forecasting (IAM4WF) that combines autoregressive and lead time embedding methods to provide accurate and reliable weather predictions. The author proposes an autoregressive training algorithm with initial frames (error free queue) for autoregressive weather forecasting models. The model is designed to address the limitations of traditional autoregressive methods in long-term weather forecasting and is expected to have potential applications in the field of climate science and weather forecasting.

**Strengths:**

In my opinion, the main contributions of this article are twofold:
1. Introducing Lead Time Embedding to the autoregressive model
2. Introduce the initial observation sequence into the model framework.

**Weaknesses:**

1. The novelty of this article is to some extent limited. Because I have to say that multi-step autoregressive training is not a particularly novel technique, as this technique has been widely used in [1,2,3].
2. The experiments on the WeatherBench dataset lack some key variables, such as T500 and Z850. I recommend the author highlighting T500, Z850, and T2M variables in the main paper. If author can supplement the multivariate experiments on the WeatherBench, it will make the experiment more complete and comprehensive.
3. At the same time, the author needs to elaborate on how many frames were extrapolated in each experiment.
4. I think the author also needs to provide a detailed description of the impact of error free queue on the experiment. I recommend the author to conduct an ablation experiment, which removes error free queue and comparing the experimental results with only error queue using algorithm 1.
5. I think the autoregressive training method of algorithm1 will consume several times more memories and computational cost than normal training. I think the author needs to conduct in-depth analysis of memories and computational cost.

Suggestions1:
This article lacks some recent NWP work:
[1]Graphcast: Learning skillful medium-range global weather forecasting.
[2]FengWu: Pushing the Skillful Global Medium-range Weather Forecast beyond 10 Days Lead
[3]FuXi: A cascade machine learning forecasting system for 15-day global weather forecast

Some suggestions about Algorithm 1: Although I can roughly guess that "e", "d", and "p" represent "encoder", "decoder" and "predictor" respectively, it would be the best to explain them in the annotation of the algorithm. If using "e", "d", and "p" at the same time, it is best to change the "encoder" in the algorithm  to "e".

**Questions:**

please check the weakness

---

### Meta-Review · Area_Chair_yToa · 2023-12-05

**Metareview:**

The paper presents the Implicit Stacked Autoregressive Model for Weather Forecasting (IAM4WF), combining autoregressive and lead-time embedding methods for weather predictions without error accumulation. The authors propose an autoregressive training algorithm with initial frames and predictions systematically queued in sequence ("error-free queue") for autoregressive prediction.

Strengths:
* Reviewers 33K8 and GNg4 praise algorithmic contributions (combining lead-time embedding for autoregressive models and initial observation sequence)
* Reviewer Dqwm thought the description and evaluation of methodology was clear.
* Promising experiments (GNg4)

Weaknesses:
* Lack of novelty, given that most weather forecasting models published in 2022-2023 used multi-step autoregressive training (33K8,Dqwm,sYk9) and limited contribution (sYk9,GNg4)
* Incomplete evaluation on WeatherBench (33K8) or on toy problems only (sYk9,GNg4)
* Missing ablations for the error-free queue (33K8)
* Expensive training (33K8)
* Missing SOTA references from 2023 (33K8,sYk9,GNg4)
* Presentation and clarity (sYk9), with confusing focus on video prediction (sYk9,Dqwm,GNg4)

The paper has scores 3, 3, 3, 5 and the authors have not provided any rebuttal. Based on the reviews, I vote to reject the paper and recommend to the authors to follow through on the suggestions made by the reviewers and to compare their methods to SOTA weather forecasting methods.

**Justification For Why Not Higher Score:**

Scores of 3, 3, 3 and 5.

**Justification For Why Not Lower Score:**

N/A

---

### Decision · Program_Chairs · 2024-01-16

Reject